# The Mushroom Glucans: Molecules of High Biological and Medicinal Importance

**DOI:** 10.3390/foods12051009

**Published:** 2023-02-27

**Authors:** János Vetter

**Affiliations:** Department of Botany, University of Veterinary Medicine Budapest, Rottenbiller 50, 1077 Budapest, Hungary; vetter.janos@univet.hu

**Keywords:** mushrooms, glucans, structural properties, synthesis, glucan determinations, glucan content, lentinan, pleuran, grifolan, schizophyllan, krestin

## Abstract

Carbohydrates, including polysaccharide macromolecules, are the main constituents of the fungal cell wall. Among these, the homo- or heteropolymeric glucan molecules are decisive, as they not only protect fungal cells but also have broad, positive biological effects on the animal and human bodies. In addition to the beneficial nutritional properties of mushrooms (mineral elements, favorable proteins, low fat and energy content, pleasant aroma, and flavor), they have a high glucan content. Folk medicine (especially in the Far East) used medicinal mushrooms based on previous experience. At the end of the 19th century, but mainly since the middle of the 20th century, progressively more scientific information has been published. Glucans from mushrooms are polysaccharides that contain sugar chains, sometimes of only one kind (glucose), sometimes having several monosaccharide units, and they have two (α and β) anomeric forms (isomers). Their molecular weights range from 10^4^ to 10^5^ Da, and rarely 10^6^ Da. X-ray diffraction studies were the first to determine the triple helix configuration of some glucans. It seems that the existence and integrity of the triple helix structure are criteria for their biological effects. Different glucans can be isolated from different mushroom species, and several glucan fractions can be obtained. The biosynthesis of glucans takes place in the cytoplasm, the processes of initiation and then chain extension take place with the help of the glucan synthase enzyme complex (EC 2.4.1.34), and the sugar units are provided by sugar donor UDPG molecules. The two methods used today for glucan determination are the enzymatic and Congo red methods. True comparisons can only be made using the same method. Congo red dye reacts with the tertiary triple helix structure, and the resulting glucan content better reflects the biological value of glucan molecules. The biological effect of β-glucan molecules is proportional to the integrity of the tertiary structure. The glucan contents of the stipe exceed the values of the caps. The glucan levels of individual fungal taxa (including varieties) differ quantitatively and qualitatively. This review presents in more detail the glucans of lentinan (from *Lentinula edodes*), pleuran (from *Pleurotus ostreatus*), grifolan (from *Grifola frondose*), schizophyllan (from *Schizophyllum commune*), and krestin (from *Trametes versicolor*), along with their main biological effects.

## 1. Introduction

More and more research and measurement data are being published on the composition and different constituents of mushrooms. Nowadays, mushrooms (both wild and cultivated taxa) have a dual role, since on the one hand they are increasingly important and valuable foods, and on the other hand, they are sources of active substances of increasing importance (i.e., molecules with biological activity) [1,2].

The wall is a very important organelle of fungal cells, having different vital functions: it is responsible for mechanical protection, osmotic conditions, protection from dehydration, binding of distinct molecules, etc. [3,4,5,6]. The cell wall has a role in permeability, in the movement of intracellular enzymes into the environment, and in the uptake of hydrolyzed metabolites from nature. The problem of understanding cell wall composition and structure has been an important mycological question for decades. A newer understanding of the systems of the fungal world (according to which fungi and mushroom-like organisms belong to Protista, Chromista, or Eumycota kingdoms) indicates that the cell wall components of each group are typically different (e.g., cellulose is not a component of the cell wall of Eukaryota fungi). The focus of interest is on the cell wall structure of large fungi; the “mushroom” category essentially refers to Ascomycetes and Basidiomycetes (the taxa with fruiting bodies). According to Grün’s model [3], the cell wall contains different glucans (α- and β-glucans), glycoproteins, and chitin. Based on Fesel and Zuccaro’s model [4], the chitin layer is located in the cell membrane, while the middle position includes the glucan molecules (α- and β-glucans), and the outermost layer consists of mannoproteins.

The cell wall components of fungi can represent a considerable part (up to 40–50%) of their body weight. It seems today that determining the in vivo structure of the cell wall will provide challenges for researchers for a long time to come. Isolation and extraction of the different cell wall constituents clearly involve the destruction of the in vivo structures, so it is difficult to deduce the structure of the original “working” cell wall.

In the case of fungi, the existence of the hypha–mycelium–fruiting body relationship is critical (i.e., that a large mushroom fruiting body is made up of a complex network of hyphae). The functional essence of the hyphal structure is the great cell wall surface with which the hyphal system is in contact with the outside world and through which its most important life processes (e.g., nutrition) are realized. 

The aim of our work is to present and characterize a group of active ingredients, mushroom glucans. The chemical structure and types of this group of molecules, their role in fungi, their distribution, and their measurable amounts in different species are discussed in many scientific publications, the number of which is growing rapidly [7].

The biological effects of glucans are investigated at a variety of research levels and methods (from in vitro studies to the effects of isolated, purified preparations to the ingestion of mushrooms as food). Other biological effects of glucans can be known mainly from folk medicine and medical traditions and partly from a variety of early scientific publications [3,8,9]. The main goal of this work is to provide guidance in the growing collection of information and to do it all from a mycological perspective.

## 2. Chemical Composition of Mushrooms: Carbohydrates

Studying the published data series on the composition of mushrooms, two important conclusions can be made regarding the total carbohydrate content:This parameter was calculated with the help of other measured parameters: total carbohydrate content = 100 − (crude protein + crude fat + ash + crude fiber) [6].The data on the dry matter basis indicate that carbohydrates account for one-half to three-quarters (possibly even more) of the dry weight of mushrooms, so they are the largest component of mushrooms. In the group of carbohydrates, a wide variety of compounds can be found, from simple sugars to complex, high molecular weight polysaccharides.

We would like to present the above in Table 1, where partly the data of the most important cultivated taxa [2] and partly the data from Kalac’s handbook [6] for wild-growing taxa are summarized. In the cases of the three most important cultivated species, the data for total carbohydrate content vary in the range from 50 to 75–85%, and are rarely higher than 90% (*Auricularia auricula-judae)* or lower than 50% (*Tuber aestivum*). The same can be established for the other part of Table 1 presenting data on wild species (genera) [6], but the data there are somewhat lower.

Therefore, carbohydrates are a group of substances that make up most of the dry mass of mushrooms. They are characterized not only by their large quantity but also by their structural diversity. This diversity is true not only in a chemical but also in a biological sense, since a simple sugar molecule can be an energy source, a building unit, and a functional element of the cell wall (chitin or glucans). Many are also characterized by the fact that they can act later as various biologically active molecules in a consuming animal or even in humans.

The grouping (overview) of the carbohydrates in mushrooms can be reviewed with the help of Figure 1. The main groups are mono-, oligo-, and polysaccharides, where one, a few, or many sugar units are located, respectively. The monosaccharides of mushrooms are sugars with 5 or 6 C atoms; free glucose, fructose, and arabinose molecules are mostly present. Their concentration is usually low or very low. Glucose, as a substrate for energy-producing processes, has a very low concentration at a given moment, and it should be mentioned that sugars are also often used for various biosynthetic processes. Monosaccharide concentrations detected and measured in large mushroom taxa are highly variable; often, the concentration can barely be detected, or it is only a few percent [6]. Among the sugar derivatives, sugar alcohols are very important, especially mannit (mannitol), whose molecules are synthesized from fructose in a two-step process. Its role in the formation and maturation of fruiting bodies is likely as an osmotic regulator, all of which is confirmed by the tendency that the mycelium < primordium < mature fruiting body indicates an increasing concentration of mannitol. The mannitol content of the mycelium is a few percent, while it rises to 20–40% in the mature fruiting body [2].

In the group of oligosaccharides, sucrose (saccharose) is rarely found. The disaccharide trehalose, consisting of two glucoses in α,α-1,1 linkages is a common component of mushroom carbohydrates, and quantities of a few percent have been measured in cultivated mushroom species (the lowest in *Agaricus bisporus*, the highest in *Pleurotus ostreatus* fruiting bodies) [6]. Regarding the function of trehalose, recent studies [10] suggest that it can play a protective role against abiotic stress conditions (i.e., heat stress processes increase trehalose content).

There are two main groups of polysaccharides: homopolysaccharides, which have the same building blocks, and heteropolysaccharides, which contain several monomers. The main groups of homopolysaccharides are chitin, glycogen, and glucans (Figure 1).

Chemically, the chitin molecule is composed of 1,4-N-acetyl-D-glucosamine monomers. Chitin is a water-insoluble, very resistant component of fungal cell walls (all fungi in the Eukaryota kingdom characteristically have chitin-containing cell walls). Our earlier studies [11,12] contained data and conclusions on the chitin content of many fungi. The cultivated button white mushroom varieties, for example, have 6–8% dry matter (DM), but the rate of chitin contents in pileus and stipes is less than 1.0% (0.8–0.9%). The oyster mushroom contains a significantly lower chitin content (2.15–5% of DM), but the pileus parts contain significantly more chitin than the stipes (chitin of pileus/chitin of stipes: 1.25–1.30). The actual chitin level of a mushroom fruit body seems to be an important factor in its digestibility because the required chitinase activity in the digestive systems of animals and humans is very low. However, the importance of chitin content in nutritional physiology also has another aspect: it forms an important part of the dietary fiber fraction and thus plays a role in ensuring the fiber requirements of normal digestive processes.

Glycogen is an interesting member of the group of homopolysaccharides. These molecules have starch-like properties (they are also called animal starch); in the case of mushrooms, they make up 5–15% of the dry matter. Glycogen contents were estimated to be between 2 and 10% (DM) in *Lentinula edodes* fruit bodies [13]. The measured concentrations were influenced by the spawn source and the characteristics of the environment during cultivation. The immature fruit bodies have lower content than the mature ones.

## 3. The Mushroom Glucans

### 3.1. Historical Background

The history of learning about glucans is logically intertwined with the ancient events and traditions of the use of mushrooms for medicinal (folk medicine) purposes. In Egypt, mushrooms are called a “gift from the god Osiris”, in Rome they are called “foods of the Gods”, and the evaluation comes from the Greek world, “elixir of the life”.

If we look at the empires of the Far East, we also have to remember their very ancient origins: the *Ganoderma lucidum* species has perhaps the longest history of medicinal use. Very characteristic mushroom names have been developed and persisted: the *G. lucidum* is known in Japan as reishi (or manetake: 10,000-year-old mushroom); in China, the same mushroom is called ling zhi (mushroom of immortality) [14]. The use of mushrooms in folk medicine was followed relatively late by scientific studies that met the basic criteria of research. The year 1957, when Lucas [15] first demonstrated the anticancer effect of mushrooms, is very notable.

Extracts of *Boletus edulis* and some other mushrooms inhibited the Sarcoma 180 cancer type in mouse tissue; the extract of the fruiting body of *Calvatia* (now: *Langermannia*) *gigantea* significantly reduced the tissue growth of several types of cancer [15]. The study of the fungi that affect the immune system developed in the 1960s and 1970s [3]. The studies, for example, started and continued in two directions. One direction (mainly in the USA, Europe, and Japan) investigated the effects of polysaccharide mixtures (zymosan) isolated from the cell wall of yeast (*Saccharomyces cerevisiae*). The second direction, which started in Japan, already demonstrated the nonspecific immunomodulating effect of β-glucans for the first time in the case of shiitake [9]. An important fact of the early studies was that the toxicity of the purified mushroom extracts was very low.

### 3.2. Structural Properties of Glucans

Glucans are the most important and abundant constituents of homopolysaccharides. Their basic unit is glucose, so at first glance, their structure is extremely monotonous; however, in reality, the molecular group is characterized by heterogeneity. This heterogeneity manifests itself, for example, in the types of bonds, the size and conformation of the molecules, the degree of branching, etc. and fundamentally affects the biological properties of the molecules as well. Next, we will review the sources, causes, and consequences of the diversity provided by the structure.

#### 3.2.1. Glycosidic Bonds

The monosaccharide units are joined by O-glycosidic linkages. Such linkages are formed from the glycosyl moiety of hemiacetal and the hydroxyl group of another monomer. What causes the diversity of the resulting polysaccharides? One factor is that the linkage of the resulting sugar chain can be (1 → 3), (1 → 4), or (1 → 6), where the numbers indicate the carbon atoms involved in the linkage. Another factor is that the resulting sugar chain can be linear or branched to varying degrees.

The existence of stereoisomers (the α- and β-configurations) known from sugar chemistry is also very important. The lowest-numbered ring carbon of a pyranose is the anomeric carbon atom. Isomers that differ only in the configuration of the anomeric carbon atom are called anomers. The α-anomer of a D-glucopyranose has an OH group pointing down axially, while the β-anomer has an OH group pointing up equatorially (Figure 2). The stereochemical difference between the α- and β-anomers—it seems—fundamentally affects the biological effect of the molecules, their nature, and their strength (see later).

#### 3.2.2. Monosaccharide Composition 

In the case of a homomeric structure, the macromolecule is made up of the same monosaccharides, while in the case of a heteromeric structure, several types of monosaccharides make up the molecule in different proportions. Generally, mushroom polysaccharides are composed of glucose, galactose, and mannose, but other sugars can also be found (e.g., arabinose, xylose, fucose, ribose) [16]. In glucans, the constituents are mostly only glucose units, although in some important glucan types, other monosaccharides are also present (the β-(1 → 3)-(1 → 6) glucan of *Grifola frondosa* contains xylose and mannose, or the pleuran of *Pleurotus ostreatus* also contains galactose and mannose).

#### 3.2.3. Backbone, Side Chains, Degree of Branching

As we saw earlier, the two main groups are linear (without branches) and branched glucans. The main chain is mostly connected by (1 → 3) (1 → 4) bonds, and the side chain(s) are connected by (1 → 6) bonds. The number of branches is well characterized by an indicator, the degree of branches (DB). For example, the DB value in lentinan is between 0.23–0.33, while it is 0.25 for pleuran, 0.31–0.36 for grifolan (from *G. frondosa*), and 0.36 for schizophyllan. It seems that for biologically active β-glucans, the DB value varies within narrow limits [3].

#### 3.2.4. Molecular Weight

An important property of glucan molecules is their molecular weight. The available data show a high degree of diversity here as well, and there are several reasons for this phenomenon. One reason is that these macromolecules from different species (types) logically differ (can differ) from each other. The other factor is that the methods of determining the molecular weight are constantly evolving and changing; moreover, the different methods are used after very different preparations (e.g., extraction, purification, etc.). This logic also includes, of course, whether there are any changes in the structure, size, etc. of the macromolecule during the application of different method combinations. Taking all of this into account, we must evaluate the available data, bearing in mind that a realistic comparison is only possible for data obtained with the same method.

Based on the work of Du and colleagues [17], we present the limits (in Da) for the molecular weight of some mushroom glucans (Table 2). The table also contains the methods used for the determinations. The determined molecular weights of the glucans isolated from *Ganoderma lucidum* are in one case on the order of 10^4^–10^5^ Da, while the values from another study are an order of magnitude higher (10^5^–10^6^ Da) (see Table 2).

The high degree of variability experienced during the determination of molecular weights is due to differences in the extraction operation and the methods of determination (i.e., the interaction of several factors). All of these must be considered when evaluating and comparing molecular weight values. However, it is a fact that (see later), according to general experience, the biological effect of glucans with a higher molecular weight is usually greater.

#### 3.2.5. Helical Conformation

The question of the possible configuration has long been a problem for glucans as well. Bluhm and Sarko (in the mid-1970s) may have been the first to publish their work on the configuration of mushroom glucans (more precisely, lentinan, which was already known at that time) [18]. The lentinan sample was obtained from Japan and examined using the X-ray diffraction method.

With the data obtained in this way and theoretical spatial structure analysis, the possibility of simple and multiple helical structures was investigated. Based on the X-ray diffraction data, a hexagonal unit was revealed, where a = b = 15.8 Å and c (fiber repeat) = 6 Å. The predicted conformation of the molecule had five structural variants: the single helix, two variants of the double helix, and two forms of the triple helix. The right-handed triple helix model is the probable configuration of the investigated lentinan molecule [18], but it is likely that other mushroom glucans (e.g., from *Armillaria mellea*) also have the same conformation’s structure.

Further research on lentinan has provided new information regarding the configuration of glucans in general. It turned out that various structural transitions and transformations can occur; the triple helix conformation can turn into another structure (e.g., into a single helix). The simple chain structure is transformed into a triple helix when lyophilized after dissolution in an 8 M urea solution and dialyzed against distilled water [19]. Lentinan has a triple helix conformation in aqueous solution, while it shows a random coil structure in dimethyl sulfoxide (DMSO) [20]. These authors used atomic force microscopy (AFM) to determine the shape of the lentinan molecule in an aqueous solution. The lentinan molecule in water exhibits wormlike, linear, circular, and crossover types.

The triple helix structure of polysaccharides in water is created and maintained by inter- and intracellular hydrogen bridges. It is also important that the distinct glucose-containing side chains cause essential differences in the physical properties of glucans. The role of hydrogen bonds created with the help of water molecules is very important in the connection between the main and side chains and between the side chains. The transformation of the configuration of glucan molecules depends significantly on certain factors, such as solvent, temperature, and pH value.

Lentinan shows a triple helix structure in aqueous solution, but it irreversibly transforms into a single strand coil in a solvent mixture where the proportion of water is only 0.15 and DMSO is 0.85 at 25 °C [21]. An irreversible helix-coil transition occurs when the concentration of NaOH is between 0.05 and 0.08 M because here the lentinan molecule is denatured [20]. Under certain experimental conditions, the triple helix structure can be reversibly restored (by dialysis on a regenerated cellulose column at 15 °C for 7 days). With the help of heat treatment (130–145 °C), an irreversible conformation change (helix → coil) can also be achieved. If increasing amounts of water are added to lentinan dissolved in DMSO, the chain collapse and aggregation process can be observed in addition to the helix → coil transition.

If the ratio of water is 0.1, the molecule exists as a random coil, if the ratio of water is 0.25, the chain shrinks, and if the ratio of water is >0.25, a connection between the collapsed chains is formed, forming quite large aggregates [21]. More recent studies [17] have indicated that the previously presented spatial conformation (triple helix) can be broken down into smaller parts for various reasons that preserve their native chemical properties and spatial structure [17,22]. Recently, more and more attention is being paid to the methodological possibilities (chemical, enzymatic, and physical methods) with which glucan molecules can be broken down into smaller parts.

Certain conformations of glucan molecules show characteristic reactions with certain dyes; for example, the single helix structure of β-(1 → 3) linked polysaccharides can be detected with aniline blue dye [23]. The triple helix structure forms a complex with Congo red dye in an alkaline solution, whereby the strong interaction between the dye and the polysaccharide molecule stabilizes. The absorption maximum of Congo red consequently shifts from 489 nm to 520 nm, all of which enable spectrophotometric quantification (see Section 3.4).

#### 3.2.6. Solubility

The solubility of an active substance is a fundamental property because during its operation and effect, it is included in such basic phenomena as stability, emulsifying ability, transport of the active substance, and membrane-forming properties. Thus, the question is: What properties of molecules affect solubility? The review work of Du and colleagues [24] draws attention to some important phenomena. If the molar mass is higher, the solubility decreases, and conversely, if the molar mass decreases for any reason, the solubility can increase significantly. An example of the latter case is when chemical modification takes place on β-glucan molecules (e.g., sulfation), which introduces ionic groups, and smaller fragments are formed from glucan, which increases solubility. Under the influence of gamma radiation, the molecular mass can also radically decrease, leading to an increase in solubility [24].

#### 3.2.7. Extraction

The extraction of mushroom polysaccharides—including various glucans—is an important issue, both in terms of determining the glucan content and isolating, purifying, and then using the molecules. The objects of glucan extraction can not only be the fruiting bodies (or parts thereof) but also the mycelium or even the nutrient medium of the fungus (cultural broth). Since most mushroom glucans are water-soluble, the main possibility is water extraction. Presently, many techniques can help in extraction procedures (ultrasonic-assisted, microwave-assisted, enzyme-assisted, subcritical water extraction, and others [25]). Conditions for optimal extraction were 94 °C, 10 h extraction time, and ratio of solid to liquid 1:6 [26]. Classical hot water extraction is a simple and feasible method, but there are several drawbacks (relatively long reaction time, high temperature, high energy demand, and relatively low extraction efficiency).

Additional problems with the extraction options are whether they lead to the degradation of glucans or to a change in the conformation of molecules, which is important from the point of view of biological effects. The large amount of literature available [8,25,26,27] suggests that the glucans of a given mushroom species should be examined specifically in light of the effects of different extraction factors (e.g., time, temperature, etc.).

### 3.3. Synthesis

The mechanism of β-glucan synthesis is a question under continuous investigation [28]. The overall process of this synthesis is partly like chitin synthesis:The β-glucan chains (approximately 1500 monomers) are produced in the cytoplasm. Later, these chains are transferred to the periplasmic space with the help of a transmembrane enzyme complex [28,29,30].The chain’s structure can be modified in the periplasmic space. For the resistant cell wall structure, β-(1 → 6)-glycosidic side branches are required, the rate of which is approximately 3–10% of the total number of glycosidic linkages. These side chains can connect several β-(1 → 3) glucan chains together [29,30]. The abovementioned processes are multi-step: the first reaction level is initiation, followed by elongation of the chain and then formation of the branches, which is a crucial step.Among the mentioned phases, the chain extension step is the most well-known, where the sugar donor uridine diphosphoglucose (UDPG) molecules deliver the new sugar unit, and the enzyme is 1,3-glucan synthase (GLS) (EC. 2.4.1.34). The process of glucan synthesis has already been investigated in more detail in the cases of some important large mushroom species (e.g., *Pleurotus, Agrocybe*/today: *Cyclocybe*/*Lentinula edodes*, *Auricularia auricula-judae*, etc.).During the reaction, the GTP molecules are activated (with the use of UTP and the formation of UDP); then, the UDPG molecules are connected to the already existing glucan chain in the enzyme complex located on the cell membrane. The enzyme complex has a hydrophilic loop responsible for catalyzing the chain extension reaction. The assumed mechanism also includes other catalytic molecules (proteins) [28].GLS enzymes known from fungi are characterized by variable substrate specificity, with an optimal pH between 5.8 and 7.8, an optimal temperature between 20 and 37 °C, and the fact that mainly divalent cations (Mg^2+^, Ca^2+^, Fe^2+^) can play a role in enzyme activity stimulation or inhibition. The value of the Michaelis constant (K_M_) of the enzyme complex seems to depend strongly on the species from which the enzyme was isolated. In summary, it can be concluded that GLS, the main enzymatic participant in glucan biosynthesis, is an enzyme that shows a high degree of heterogeneity [28].

### 3.4. Determination of Glucan Contents

In the first decade of our century—due to the increased interest in beta-glucans—it became increasingly necessary to develop and compare determination methods.

#### 3.4.1. Enzymatic Method

Bak and colleagues [31] applied the so-called enzymatic method. The authors used an enzyme kit (Megazyme, Ireland), which consists of exo-1,3-β-glucanase, β-glucosidase, amyloglycosidase, and invertase enzymes, as well as components necessary for glucose determination (glucose-oxidase, peroxidase, and 4-aminoantipyrine).

During the measurement of the total glucan content, the mushroom samples were hydrolyzed in 37% HCl for 45 min at 30 °C and then at 100 °C for 2 h. After neutralization, the glucose was hydrolyzed with 1,3-β-glucanase and β-glucosidase in Na-acetate buffer (pH = 5) for 1 h at 40 °C, and then the absorbance of the solution was measured at 510 nm. The alpha-glucan content was determined after hydrochloric acid hydrolysis with other enzymes: amyloglucosidase and invertase. The β-glucan content was calculated: β-glucan = total glucan−α-glucan content. The authors performed the enzyme method combination on mushroom mycelium, fruiting body cap, and stipe samples in 10 shiitake varieties.

Sari and his workgroup analyzed [32] many cultivated and wild mushrooms, mainly in fractionated form for caps and stipes. Their method was the enzymatic analysis described above, but the α-glucan content was measured after hydrolysis in 2 N KOH solution (based on the advice of the enzyme manufacturer). 

#### 3.4.2. Congo Red Method

In 2011, Mölleken developed a method that is based on the reaction between Congo red dye and glucans with a triple helix structure, so it is specific for β-1,3-1,6 glucans, which have a spatial structure that is suitable for determining their quantity in each mushroom or in mushroom preparations [33]. The enzymatic (Megazyme) and Congo red methods were used in parallel to compare the β-glucan content of 18 wild and three cultivated mushroom species, which is also suitable for comparing the methods [34]. The meta-analysis of their data points out (Table 3) that there is a significant difference between the data sets of the two methods; the β-glucan content obtained using enzymatic analysis is on average two and a half times higher than the data obtained using the Congo red method.

If the samples of wild mushrooms are compared separately, the smallest difference is found in *Lactarius deliciosus* (1.61 times), while the largest is defined in *Cantharellus cibarius* (the data obtained using the enzyme method is 27.6% of DM, while the data obtained using the Congo red method is only 1.94%, i.e., the difference is 13.3 times based on the data of [34]). For the three cultivated mushroom species (Table 4), the enzyme method indicates a more than three times higher amount of glucan (for *Agaricus bisporus*, *Pleurotus ostreatus*, and *Lentinula edodes*, the differences are 3.64, 3.25, and 3.12 times higher, respectively) [34]. The above data emphatically underline that the comparison of the glucan content of mushrooms is only possible with data sets obtained by the same method. The importance of the question and the interpretation of the data are also great because the real comparison and standardization of mushrooms, mushroom preparations, and food supplements require a uniform methodological background.

### 3.5. Glucan Contents of Mushrooms

The data on the α-, β-, and total glucan contents of mushrooms are quite limited (compare with what was explained in Section 3.4), so their evaluation requires attention. Data from Sari [32] provide an opportunity to evaluate and meta-analyze the glucan contents of the caps and stems of the studied wild and cultivated species (Table 5 and Table 6).

According to the summary of the data from the wild species, the total glucan content of the caps was lower than that measured in the stipes (cap average: 22.07% DM, stipe average: 29.50% DM), while the α-glucan content showed no clear difference between the caps and the stipes.

The minimum and maximum values of the glucan content showed a very wide range, which is especially true for the values measured in the stipes. If we examine the original data table, according to the examined species, we also find one or two outstanding and interesting data. For example, the very low glucan content of *Xerocomellus chrysentheron* or the fact that the stipe of *Boletus edulis* has very high glucan parameters (total glucan: 63.3% DM, β-glucan: 57.3% DM).

We worked out (based on [32]) the proportion of the glucan fractions of caps and stipes (Table 7). The α-glucan fractions occurred in essentially the same amount in the stipe and the cap (ratio: 0.91 and 0.98); for β- and total glucan contents, the stipes contained approximately 30–40% more glucans than the caps. The fruiting bodies of the wild taxa and the three cultivated mushrooms had essentially the same proportions.

A recent publication from Ciric examined the β-glucan content of food supplements containing different mushrooms, using the previously mentioned and presented enzymatic method [35]. The average β-glucan content of the 10 tested types of food supplement capsules was 18.45% DM; the lowest measured value was 5.5%, and the highest glucan content was 37.5% DM (the latter capsule contained *Ganoderma lucidum* powder and shiitake extract powder).

In the previously cited work by Bak [31], the α-, β-, and total glucan content of 10 varieties of *Lentinula edodes* (shiitake) were examined in the mycelia, cap, and stipe of the varieties using the already presented enzymatic (Megazyme) method.

On average, shiitake varieties contain a significant amount of total and β-glucan in the cap (39.1% and 35.9%, respectively). The mycelia of the varieties have a significantly lower glucan content, but here, the amount of α-glucan is relatively the highest (total: 27.4%; β-glucan: 22.2%). The reported data are also interesting because they indicate that not only the two parts of the fruiting body—the cap and the stipe—but also the parts of the mycelium and fruiting body are significantly different. It seems that the amount of glucan increases significantly during the development of the fruiting body, since the ratio of total glucan contents for mycelium:cap:stipe = 100:161.7:199.6.

The α- and β-glucan contents of mushrooms can be influenced by different factors, such as the drying and blanching processes. The fruiting bodies of shiitake mushrooms were dried at 35, 45, and 55 °C, and drying at 60 °C already reduced the amount of the soluble fraction. The drying temperature of 55 °C was found to be optimal because it had the shortest drying time [36].

The process of blanching is important among preservation treatments, as it inactivates decomposing enzymes, pre-shrinks, removes air, reduces oxidative changes, differentiates microbial populations, reduces cooking time, etc. [37]. During an examination of 11 Thai mushroom species, the authors concluded that the blanching process generally decreased α-glucan while increasing β-glucan.

### 3.6. Presentation of the Most Important Glucans

#### 3.6.1. Lentinan 

The source of lentinan is the fruiting body and mycelium of the *Lentinula edodes* (shiitake) mushroom. It was first isolated by Chihara’s group in 1970 [38]. The sugar chain of glucan consists of (1 → 3)-β-D-glucopyranoside units, and two (1 → 6)-β-D-glucopyranoside side chains are attached to every five such sugar units (Figure 3). The degree of branching (DB) value of the molecule is between 0.33 and 0.5. The molecular weight of lentinan was initially measured as 9.5–10.5 × 10^5^, and then with another method, it was found to be lower, in the range of 3–8 × 10^5^ [20]. 

The molecular weight of the glucan fractions from shiitake—according to recent data—varies between 300 and 800 kDa, with an average of 500 kDa. The configuration of the molecule is as follows: in aqueous solution, the H-bridges form a triple helix structure, while the configuration is a random coil when dissolved in DMSO [20].

Among the diverse and multifaceted biological effects of lentinan, the anticarcinogenic (antitumor) and immune system stimulating and activating effects are decisive. The first scientific document about its anticarcinogenic effect [38] was published on the effect against Sarcoma 180 (although a series of folk medicine and historical records indicated the possibility of an antitumor effect, especially in Far Eastern societies).

The literature of the last few decades has reported somewhat incomprehensive detail and investigation (not only of lentinan but also of other mushroom glucans). Pre- and postoperative therapy with lentinan can be effective against cancer recurrence and metastases after surgical treatment [20]. The variety of data and observations logically raises questions about the mechanism of the effect. The discussion of the question focuses on several possibilities:Does direct inhibition of the growth of cancer cells occur (i.e., is there a direct anticarcinogenic effect)?Does the molecule exert its effect indirectly, through stimulation of the immune system?Is there a preventive effect in relation to the spread and migration of cancer in the body? [39]

In general, the activation of immune cells is the most important step in the indirect effect of lentinan. Studies indicate that under the influence of lentinan, cells release cytokines that serve as signal messengers. The increase in cytokine production by immune cells has been investigated in animals and humans [24]. Data suggest that lentinan can increase the ability of certain immune cells to slow down or destroy cancer cells in humans. As a result of lentinan treatment, more nitric oxide is produced, which stimulates the immune system.

Immune activation capacity can also be related to regulation by hormonal factors. In summary, lentinan can suppress the growth of cancer cells or even kill cells directly through various pathways of immune system activation [20]. In the literature, several authors have created complex models of the antitumor effects of lentinan (see the relevant literature [40]). In the case of lentinan (and other mushroom polysaccharides), much attention has been paid for a long time to the possibility and nature of the correlations between the physico-chemical properties and biological effects of the molecules. If, for example, the triple helix configuration of the lentinan changes or collapses, the molecule partially denatures, and the anticarcinogenic effect decreases. In general, there is a connection between the biological effects and the integrity of the molecular structure. Substances with higher molecular weights are generally more effective than smaller ones.

Beta-glucans entering the human gastrointestinal tract (including lentinan) show good resistance to gastric juice. They enter the small intestine unchanged, where they bind to macrophage receptors in the intestinal wall and are then transported to the spleen, lymph nodes, and bone marrow [41]. In macrophages, the large β-glucans are broken down into smaller molecules, which can bind to the complement receptors of immune cells. This ultimately strengthens the immune response against tumor cells.

The interactions between β-glucans and immune cells are very complicated, and not all connections are known. However, it is known that the common receptor Dectin-1 is activated and then the amount of reactive oxygen species (ROS), which acts against pathogenic microorganisms, increases. The receptor stimulates the production of various cytokines [42]. The mechanism of action of glucans includes additional receptors (TLRs/Toll-like receptors), complement receptor type 3 (CR3), and the scavenger receptor (Src). The essence of the effect of glucans can therefore be described by the fact that they activate cellular and humoral responses of the immune system through different receptors.

Regarding the antitumor and cytotoxic effects of glucans, the mechanism remains partially unexplored. According to the latest literature and sources [41], it has not been proven that glucans exert a direct cytotoxic or apoptotic effect against cancer cells. Although one publication refers to the direct cytotoxic activity of some glucans (e.g., in the case of liver cells [43]), recently the more likely assumption is that glucans exert their antitumor (usually anticancer) effect through the immune system.

#### 3.6.2. Pleuran 

This molecule was isolated in the early 1990s from *Pleurotus ostreatus* (oyster mushroom), noting that bioactive glucans have also been isolated from many other species of the *Pleurotus* genus (*P. eryngii*, *P. florida*, *P. pulmonarius*, *P. tuber-regium*, *P. sajor-caju*, and *P*. *ostreatoroseus*) [44]. Glucose molecules are bound with (1 → 3)-β-bonds in the chain, and every four molecules are connected to glucose with a (1 → 6)-β-bond. The molecular weight of pleuran ranges between 600 and 700 kDa [45]. Among the biological effects of pleuran are the antitumor effect, the reduction of blood lipid level, and the stabilization of carbohydrate homeostasis. It has also been shown to have antifungal properties, an increase in antioxidant potential, and an anti-inflammatory effect [45].

The clinical and immunomodulatory effects of pleuran were investigated by Urbancikova and co-workers [46]. Active treatment with pleuran caused a significantly shorter duration of herpes simplex virus (HSV-1) symptoms. The severity and length of respiratory symptoms were lowered in the treated patient group compared to the placebo group. No side effects were observed during the clinical experiments. The results suggest that pleuran is a suitable molecule for the treatment of acute HSV-1 infection. The respiratory tract symptoms changed very favorably.

In recent work by the above-mentioned authors [47], it has been confirmed that pleuran application is effective in preventing respiratory infections (recurrent respiratory tract infections/RRTIs). This method is suitable as a strategy for improving immune functions in young populations.

#### 3.6.3. Bioactive Molecules from *Grifola frondosa* (Maitake) 

The habitats of this mushroom are the temperate forests of Europe, Asia, and North America. This species has different English names: hen-of-the-woods, king of mushrooms, sheep’s head, etc. *Grifola frondosa* is a delicious mushroom, with a pleasant, sweet smell, which can be explained mainly by the high content of trehalose, glutamine, aspartic acid, and 5’-nucleotides. In addition to its nutritional value, the mushroom also has a variety of pharmacological effects. The antitumor effect of extracts made from the fruiting body with hot water was already demonstrated in the 1980s, and the β-glucans of the mushroom are the main factors in this effect. In the last 30 years, 47 bioactive polysaccharide fractions have been isolated and purified from the fruiting body, mycelium, or cultivation medium of this mushroom [48].

The mushroom contains 3.8% water-soluble polysaccharides, of which 13% is (1 → 3) (1 → 6)-β-D-glucan. In addition, heteroglucans and heteroglucan–protein complexes are also present. The so-called D (or its purified version is MD) fraction and grifolan are perhaps the most well-known fractions, but Wu’s work draws attention to the tabular presentation of nearly 31 different bioactive molecules. β-glucan from fraction D contains (1 → 6) chains with (1 → 3) side chains and (1 → 3) chains with (1 → 6) side chains. Gel-forming grifolan (GRN) molecules have immunomodulatory effects.

The antitumor effect of the mushroom was first described by Miyazaki and his colleagues forty years ago against the Sarcoma 180 cancer type in mice [49]. The experiences of the past 30 years have led to the fact that the main possibilities of the mushroom’s anticancer effect can be formulated as follows: protection of healthy cells, protection against tumor metastases, and inhibition of tumor growth. In other words, the direct and indirect effects of the mushroom against tumors occur through the stimulation of the immune system.

Today, it is also known that the D-glucan fraction of maitake can be used orally, intravenously, and intraperitoneally (many other antitumor polysaccharides are ineffective when used orally). According to experiments conducted with the D-fraction of maitake [50], it is effective in cases of breast, liver, and lung cancers but less effective in other cancers.

According to a recent summary by Wu [48], the immune-modifying effect of maitake was confirmed by several tests. The mushroom’s bioactive substances act on macrophages, cytotoxic T cells, and natural killer (NK) cells and stimulate cytokines and other signaling molecules (interferons (IFN), interleukins (IL), and tumor necrosis factors (TNF)). Biological influences include antiviral and antibacterial effects (hepatitis B, enterovirus 71, the herpes simplex virus (HSV-1), and the HIV virus).

In connection with the almost epidemic spread of diabetes, it is worth drawing attention to the effects of maitake on sugar metabolism. This effect is partly related to the activity of insulin and partly to the inhibition of α-glycosidase activity; the latter process slows down the hydrolysis of starch and thus lowers the blood sugar level [48].

#### 3.6.4. Schizophyllan (from *Schizophyllum commune*)

This glucan was first isolated from a non-edible, plant-parasitic fungus, *Schizophyllum commune* (split-gill mushroom), by Kikumoto and co-workers [51]. This white-rot mushroom grows on dead or decaying wood and is distributed on all continents (except Antarctica).

The main chain of the glucan monomer consists of three (1 → 3) linked glucoses, and a glucose unit is connected to the middle sugar by (1 → 6) bonds (Figure 4). The molecular mass is between 100 and 200 kDa, although some sources mention a significantly higher molecular weight (4300 kDa) [52]. The molecule shows a triple helix conformation in aqueous solution, similar to lentinan [51]. Schizophyllan is a water-soluble and extracellular polysaccharide. For its production, different strains of *Schizophyllum commune* are used that have different wastes of lignocellulose character, such as rice hull hydrolysate, corn fiber, date syrup, carboxymethyl-cellulose, and different sugars (e.g., glucose, sucrose, etc.) [53].

Schizophyllan was found to control the growth of Sarcoma 180 tumors. It was tested and used (mainly in Japan) against head and neck cancer, resulting in improved patient survival [53]. Different clinical trials have been performed in Japan. These studies combined schizophyllan with conventional chemotherapy (tegafur, 5-fluoroacid, and other molecules) and were applied to 367 patients with recurrent, inoperable gastric cancer. An increase in the survival rate was demonstrated [54]. In other studies, longer overall survival rates of head and neck cancer-related patients were achieved and documented [55]. Positive results were reported in another randomized, clinical trial (following surgery, radiotherapy, chemotherapy, and schizophyllan in various combinations) [56]. The above results indicate that the anticancer effect of schizophyllan exoglucan appears to be achieved in “synergism” with traditional, classic treatments. 

The biological activity of schizophyllan can be changed and improved by ultrasonic treatment of the molecule [57]. Ultrasonic-treated schizophyllan caused an increase in nitric oxide production by macrophages and enhancement of the proliferation rate of lymphocytes. This represents a completely new and different application of schizophyllan when recently a composite of the molecule and silver nanoparticles was prepared [58]. This composite has been found to be a new potential possibility for certain biomedical applications (mainly for wound healing). The silver nanoparticles, binding with schizophyllan (through a non-covalent but strong linkage), can lead to good dispersion of silver particles within the matrix of the biopolymer [58].

#### 3.6.5. Krestin (Crestin, Polysaccharide-K, PSK)

The krestin heteroglucan was isolated from the fruiting body of *Trametes (Coriolus) versicolor* (turkey tail). Since the glucan structure is connected to proteins, it is a proteoglucan (protein content between 25 and 38%) [41,45]. Its molecular weight is approximately 100 kDa. Its main sugar component is glucose, but it also contains small amounts of mannose, fucose, galactose, and xylose [59]. There are (1 → 3) bonds between the units of the glucan backbone, a side chain is connected to every fourth glucose unit by (1 → 6) bonds, and the protein parts are covalently linked to (1 → 6) side chains.

The mushroom was discovered for modern science and medicine in 1965. PSK (Poly-Saccharide-Kurahe = krestin) was discovered and patented in 1969 and commercialized in Japan in 1977. Chemical tests measured a high glucan content (61%) in the mushroom, 99.3% of which was β-glucan [29].

The effects of *Trametes* glucans: According to in vitro studies, the extracts of the mushroom, PSK and PSP (another isolated proteoglucan: Poly Saccharo Peptide), have cytotoxic effects against tumor cell lines. The mortality of patients suffering from various types of cancer improved significantly, and a relationship between survival without deterioration and *Trametes* preparations was found. PSK, for example, has been proven to be effective in the treatment of stomach, esophageal, colon, rectal, and lung cancer in doses of 3–6 g per day for 1 year, according to clinical trials [60]. PSK is usually added to chemotherapy after surgery; a comprehensive scientific overview was given by Kidd [61]. The proteoglycan isolated from *Trametes* (PSP) significantly reduced various side effects in cancer patients (e.g., loss of appetite, weakness, dry mouth, strong heartbeat, insomnia, short, rapid breathing, malaise, vomiting, and night sweats). In other words, the number of patients with no side effects increased significantly [62]. Krestin has also successfully been used in veterinary science for adenosarcoma, fibrosarcoma, mastocytoma, melanoma, mammary cancer, colon cancer, and lung cancer [63,64].

### 3.7. Relationships between Structure and Biological Effects

Glucans are physiologically active compounds; they are called biological response modifiers (BRMs). Thanks to the properties of BRMs, they can often act as remedies or adjuvants, for example, in bacterial or viral infections, or they can also be effective against tumors [65]. The structure of effective β-glucans usually means a (1 → 3)-β-D-glucose chain molecule in which some glucose units are randomly connected with (1 → 6) β-bonds. Molecules with significant antitumor activity have a degree of branching (DB) value between 0.2–0.3 (lentinan, schizophyllan), while molecules with a lower or higher DB value are less effective. In the case of native β-D-glucans, the sugar chain has a triple helix structure, parts of which can be combined with simple or double chains.

The antitumor effect of schizophyllan is characterized by a triplex helix structure and a molecular weight greater than 100 kDa. The triple helix (created by three H bonds at the C-2 position) configuration appears to exist only in glucans of a higher molecular weight. Around and below 25–40 kDa, the molecule exists only as simple fibers (chains) in aqueous solution [7]. There are also contrary opinions on the relationship between structure and effect. Some data, for example, also describe a significant antitumor effect in the case of glucans with a small molecule and a non-branched structure [66]. Therefore, the question of structure–effect correlations can still be considered open today, which clearly warrants further research.

## 4. Conclusions

This review summarizes the mycological, chemical, and biological aspects of mushroom glucans. The glucans (the homo- or heteroglucans) are significant components of the fungal cell wall. They are composed of sugar (mostly glucose) chains, where typical glycosidic bonds form the linear or even branched structure. The special configuration of β-glucans and the integrity of this triple helix structure seem essential for the molecule’s biological effects.

Among the main biological impacts of glucans are antitumor (anticarcinogenic) and immune-stimulating effects. Some types of glucans can also be associated with other groups of effects (e.g., antidiabetic, antioxidant properties, etc.). From a mycological point of view, it is important that not only the fruiting bodies but also the mycelium and even the growing medium containing the fungus can be utilized and can be sources of glucans.

Effective glucan content is (can be) present not only in edible species but also in inedible taxa (*Ganoderma lucidum, Schizophyllum commune*) as well, and they can also be used and even cultivated as raw materials to produce glucans. In several fields of medicine (e.g., oncology and immunology), the importance and possibilities of mushroom glucans are very promising.

What tasks can be set for researchers in the field for the future?

It would be necessary to clarify, develop, and standardize the determination methods available today (measuring the glucan content of existing preparations using a clear method and, at the same time, standardizing the preparations).Laboratory and then industrial application methods for the extraction and purification of different glucan types should be investigated. The produced molecules can be used in laboratories and then in clinical experiments.Most glucans can also be involved in the prevention of diseases, which, of course, also requires an attitude toward development.The solution of all these tasks requires cooperative interdisciplinary research.

## Figures and Tables

**Figure 1 foods-12-01009-f001:**
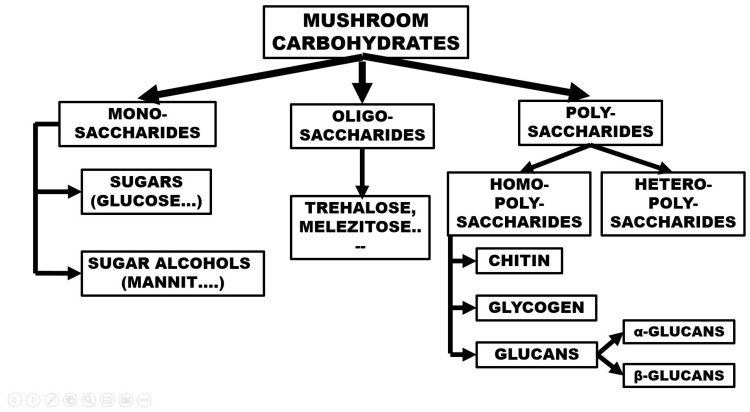
The groups of mushroom polysaccharides.

**Figure 2 foods-12-01009-f002:**
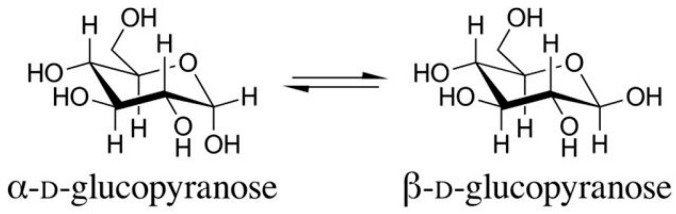
The structures of the α- and β-D-glucopyranose isomers (anomers).

**Figure 3 foods-12-01009-f003:**
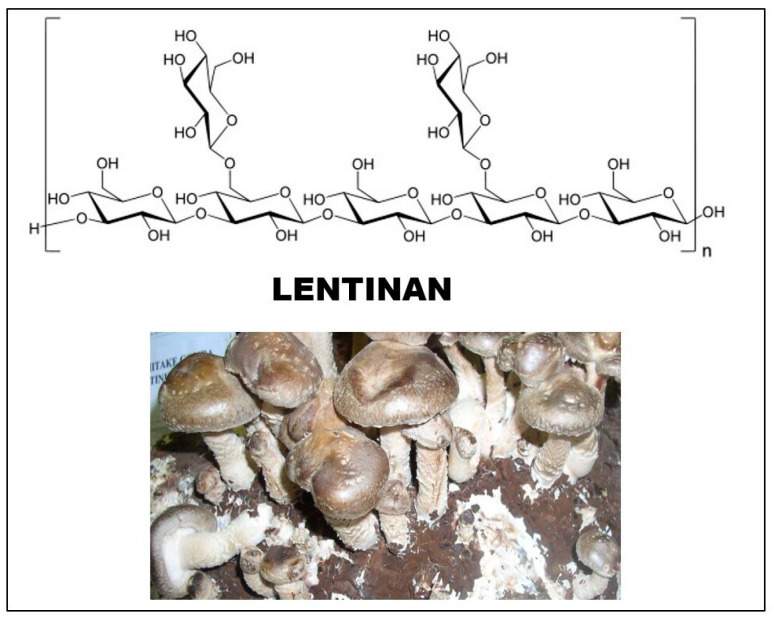
Structure of lentinan, the main glucan of *Lentinula edodes* (shiitake).

**Figure 4 foods-12-01009-f004:**
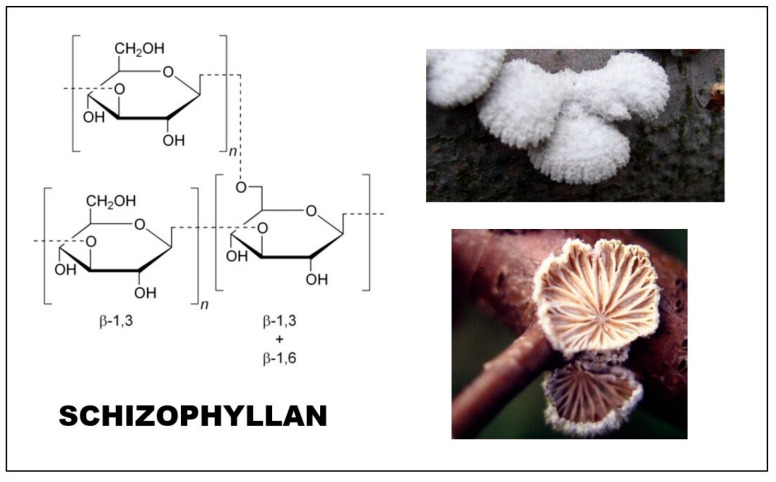
The schizophyllan glucan from *Schizophyllum commune* (split gill mushroom).

**Table 1 foods-12-01009-t001:** Total carbohydrate contents of some cultivated and wild-growing mushroom taxa [2,6].

Cultivated Mushrooms	Carbohydrate Content (% of DM) *	Wild Growing Taxa	Carbohydrate Content (% of DM)
*Agaricus bisporus* white button mushroom	50.9–74.0	*Agaricus* species	37.5–65.0
*Agaricus subrufescens* almond mushroom	39.0–64.0	*Amanita* species	49.1–72.0
*Pleurotus ostreatus* oyster mushroom	51.9–85.2	*Armillaria mellea* honey fungus	63.0–80.2
*Pleurotus eryngii* king trumpet mushroom	70.5–81.4	*Boletus edulis* penny bun	55.0–70.9
*Lentinula edodes* shiitake	67.1–87.0	*Cantharellus cibarius* golden chantarelle	31.3–72.0
*Flammulina velutipes* golden needle mushroom	56.6–86.2	*Coprinus comatus* shaggy ink cap	49.4–76.3
*Coprinus comatus* shaggy ink cap	76.5	*Flammulina velutipes* golden needle mushroom	70.9
*Auricularia auricula-judae* jelly ear	77.2–91.0	*Lactarius deliciosus* saffron milk cap	61.0–77.2
*Volvariella volvacea* straw mushroom	52.3	*Lepista (Clitocybe) nuda* wood blewit	65.9–71.0
*Ganoderma lucidum* reishi, lingzhi, lacquered bracket fungus	82.3	*Macrolepiota procera* parasol mushroom	70.3–81.0
*Tuber aestivum* summer truffle	48.9	*Pleurotus ostreatus* oyster mushroom	65.4–70.6

* Dry matter = DM.

**Table 2 foods-12-01009-t002:** Molecular weights of different glucans determined by several methods (or by a combination of methods) [16,17].

Source of β-Glucan	Methods of Determination *	Molecular Weight (Da)
*Saccharomyces cerevisiae*	SEC-RI	2.79 × 10^4^–21.75 × 10^5^
*Schizophyllum commune*	HPLC-MALLS-RI	8.8 × 10^5^–2.4 × 10^6^
*Schizophyllum commune*	HPLC-RI	1.97 × 10^5^–2.9 × 10^6^
*Schizophyllum commune*	HPLC-RI	2.9 × 10^6^
*Ganoderma lucidum*	HPSEC-MALLS-RI-VS	2.9 × 10^5^–2.42 × 10^6^
*Ganoderma lucidum*	SEC-LLS-RI	5.7 × 10^4^–4.45 × 10^5^
*Lentinus velutinus*	HPGPC	3.36 × 10^5^
*Pleurotus ostreatus*	GPC	3.3 × 10^4^
*Pleurotus djamor*	NMR	1.61 × 10^5^
*Agaricus bisporus*	HPGPC	7.84 × 10^5^

* HPLC: high performance liquid chromatography; MALLS: multiangle laser light scattering method; RI: refractive index detector; SEC: size-exclusion chromatography; HPSEC: high performance size exclusion chromatography; HPGPC: high performance gel permeation chromatography; GPV: gel permeation chromatography; NMR: nuclear magnetic resonance; LLS: laser light scattering; VS: viscosity detector.

**Table 3 foods-12-01009-t003:** β-glucan contents of 18 wild-growing mushrooms determined using the Megazyme and Congo red methods [31].

Analyzed Samples and Method of Analyses	Arithmetical Mean % of DM ± SD	Min–Max Value % of DM
β-glucan content of 18 wild-growing mushrooms assayed using the Megazyme method	24.81 ± 5.98	10.50–34.97
β-glucan content of 18 wild-growing mushrooms assayed using the Congo red method	10.03 ± 4.47	1.95–17.10

**Table 4 foods-12-01009-t004:** β-glucan contents of three commercial cultivated mushrooms species analyzed by two methods [31].

Mushroom	β-Glucan with Megazyme Method (% DM)	β-Glucan with Congo Red Method (% DM)	The Rate of the Found Glucan Contents
*Agaricus bisporus* white button mushroom	11.36 ± 2.85	3.11 ± 0.10	3.64
*Pleurotus ostreatus* oyster mushroom	40.34 ± 3.23	12.39 ± 1.10	3.25
*Lentinula edodes* shiitake	26.26 ± 3.23	8.42 ± 0.63	3.12

**Table 5 foods-12-01009-t005:** Average glucan (all, α, and β) contents in caps and stipes of 23 mushroom samples and the distribution of data (minimum and maximum values based on data of [32]).

Mushroom Samples (*n* = 23)	Arithmetical Mean (% of DM ± SD)	Min.–Max. Values (% of DM)
Cap—All glucan content (Pileus)—α-glucan content β-glucan content	22.07 ± 6.60 3.35 ± 3.26 18.29 ± 6.09	9.90–34.1 0.65–14.9 7.08–33.5
Stipe—All glucan content α-glucan content β-glucan content	29.50 ± 11.78 3.05 ± 2.68 26.38 ± 11.30	14.33–63.31 0.45–12.1 13.09–57.90

**Table 6 foods-12-01009-t006:** Average glucan (all, α, and β) contents in caps and stipes of three cultivated taxa (in % of DM) [32].

Cultivated Species	All Glucans		α-Glucan Content		β-Glucan Content	
	Cap	Stipe	Cap	Stipe	Cap	Stipe
*Agaricus bisporus*—white—average ± SD	10.05 ± 2.22	14.96 ± 4.9	1.54 ± 0.38	2.66 ± 1.22	8.68 ± 2.37	12.29 ± 4.07
*Agaricus bisporus*—brown—average ± SD	12.34 ± 4.5	14.64 ± 4.87	3.51 ± 2.38	4.15 ± 2.84	8.83 ± 3.04	10.07 ± 2.23
*Lentinula edodes*—shiitake—average ± SD	20.5 ± 5.95	26.74 ± 3.95	0.76 ± 0.40	1.44 ± 0.85	19.77 ± 6.23	25.30 ± 4.38
All cultivated taxa—average ± SD	14.40 ± 5.32	18.78 ± 6.89	2.79 ± 2.05	2.75 ± 1.35	12.42 ± 6.35	15.88 ± 8.22

**Table 7 foods-12-01009-t007:** Glucan rates from stipes and caps in wild-growing (*n* = 23) and three cultivated mushrooms, analyzed on data of [32].

Wild-Growing Mushrooms	Rate	Cultivated Mushrooms	Rate
α-glucans_stipe_/α-glucans_cap_	0.91	α-glucans_stipe_/α-glucans_cap_	0.98
β-glucans_stipe_/β-glucans_cap_	1.33	β-glucans_stipe_/β-glucans_cap_	1.27
All glucans_stipe_/all glucans_cap_	1.44	All glucans_stipe_/all glucans_cap_	1.30

## Data Availability

This publication is a review article.

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
