# Peer review of "The Mushroom Glucans: Molecules of High Biological and Medicinal Importance"

_foods, 2023, doi:10.3390/foods12051009_

Round 1
Reviewer 1 Report
The present submitted review manuscript entitled as “the mushroom glucans: molecules of high biological and medicinal importance”, is organized by Prof. Vetter, according his studies and other scientists studies in editable fungal polysaccharide.
Some comments are listed as following:
Abstract:
1. The paragraph “ this review focuses on great mushrooms,….Tremetes”. might put it up at the end of abstract section and combine with the last sentence of the abstract.
2. UDPG: into “sugar donor UDPG”
English editing and identical references part
Paragraph 3.2: structural properties of glucans: not very clear explanation?
Line 142-144: what is the meaning of pileus/stipe ratio ??
contains; (rate of pileus/stipes: 1.25-1.30)
Lines 427-430: not clear in meaning
Paragraph 3.3: Synthesis…lines 327-328, clear explanation
Correction:
Line 53: fungi, into fungi
Line 56: into Eucaryota fungi, etc
Line 61: a and a glucans, what is?
Line 128/129: trehalose, consisting of two glucose in a,a, 1-1-linkage;
Line 135: several monomers
Line 137: are composed of 1,4-N-acetyl-D-glucosamine monomer
Line 141: percent
Line 173: 1960s and ’70s
Line 187: italic “O”-glycosidic linkage
Line 302: beta-(1→3)-linked
Line 334: spelling??
Lines 361, 509: a- into Alpha; b- into Beta-
Lines 460, 530, 539, 580,: italic species name: Lentinula edodes; .. Pleurotus ostreatus
Line 622: (1→3)-b-D-glucose
Tables 1 and 4: for common name of fungi, the first letter into small letter, e.g. White button mushroom into white button mushroom, make table clearer.
Table 6: italic species name
Author Response
Rev. 1
The present submitted review manuscript entitled as “the mushroom glucans: molecules of high biological and medicinal importance”, is organized by Prof. Vetter, according his studies and other scientists studies in editable fungal polysaccharide.
Some comments are listed as following:
Abstract:
- The paragraph “ this review focuses on great mushrooms,….Tremetes”. might put it up at the end of abstract section and combine with the last sentence of the abstract.
Reflections: The sentence in question was modified and moved to the end of the abstract.
- UDPG: into “sugar donor UDPG”
Reflections: Okay, it's done.
- Paragraph 3.2: structural properties of glucans: not very clear explanation?
Reflections: I modified the paragraph slightly.
- Line 142-144: what is the meaning of pileus/stipe ratio ?? contains; (rate of pileus/stipes: 1.25-1.30)
Reflections: I formulated the concept of the pileus/stipe ratio clearly, since it is nothing more than the ratio of the given glucan measured in pileus and stipe.
- Lines 427-430: not clear in meaning.
Reflections: The paragraph deals with the summary analytical tests of various food supplements, I don't see any justification for the changes.
- Paragraph 3.3: Synthesis…lines 327-328, clear explanation.
Reflections: A small correction was made.
Corrections
- Line 53: fungi, into fungi
and
- Line 56: into Eucaryota fungi, etc.
Reflections: Today, the taxa of the fungal world in a broader sense are divided into three kingdoms (Protista, Chromista and Eucaryota). The mushroom category refers to large fruiting fungi that belong to the ascomycetous and basidiomycetous phylums of the Eucaryota kingdom. I think the information is clear.
- Line 61: a and a glucans, what is?
Reflections: Correction into α and β was made.
- Line 128/129: trehalose, consisting of two glucose in a, a, 1-1-linkage;
Reflections: The correction was made.
- Line 135: several monomers
Reflections: The plural has been marked (s).
- Line 137: are composed of 1,4-N-acetyl-D-glucosamine monomer
Reflections: Correction of glucoseamine into glucosamine was made.
- Line 141: percent
Reflections: The proposed one-word writing has been done.
- Line 173: 1960s and ’70s
Reflections: The correction”1960s and 1970s has been done.
- Line 187: italic “O”-glycosidic linkage
Reflections: I don't really know the reason for the requested italics.
- Line 302: beta-(1→3)-linked
Reflections: It was corrected into: β-(1→3)-linked
- Line 334: spelling??
Reflections: Unnecessary word deleted.
- Lines 361, 509: a- into Alpha; b- into Beta-
Reflections: Position at the beginning of a sentence, so I corrected it for the words alpha and beta.
- Lines 460, 530, 539, 580,: italic species name: Lentinula edodes; .. Pleurotus ostreatus
Reflection: The requested corrections have been made.
- Line 622: (1→3)-b-D-glucose
Reflection: (1→3)-β-D-glucose correction has been made.
- Tables 1 and 4: for common name of fungi, the first letter into small letter, e.g. White button mushroom into white button mushroom, make table clearer.
Reflections: The requested corrections have been implemented.
- Table 6: italic species name
Reflection: Correction has been made.
Submission Date 13 January 2023
Date of this review 30 Jan 2023 07:08:27
General comments for all reviewers:
- I tried to correct the text based on aspects of linguistic correctness (in some places, I condense it, I broke several long sentences into parts). I tried to avoid, for example, repetition of words by using synonymous words.
- Due to the insertions and the use of a larger font size, the line numbers of review have changed therefore the number of lines given in the reviewer's opinion are no longer valid.
Finally, I ask you to approve the implemented modifications and corrections.

Reviewer 2 Report
The review presents relevant information, it is coherently organized and emphasized the most relevant that has been published on glycans. The author mentions the polysaccharide of some mushrooms, however, the information he puts on pleurotan and schizophyllan is scant compared to the others (lentinan, maitake, krestin).
In general, I suggest reviewing the format, some comments about it:
• In the text there is a font of different sizes (lines 7-14), the information in section 3.3 is essential, I don't see why putting smaller fonts (lines 323-349)
• In several parts of the text the author opens parentheses and they do not close, you should consider putting brackets in the places where there will be double parentheses
• In the words that you want to abbreviate, you usually put complete and in parenthesis the abbreviation (line 141), and in line 153, put the abbreviation in parentheses, review and correct where applicable.
• Several paragraphs are not justified
• The general format of the writing is to put a title and give a space, it does not do so in section 3.6.1 (line 460)
• Some scientific names are not in italics.
The information about the glucans of Schizophyllum and Pleurotus presented is little, compared to that presented for Grifola frondosa, Trametes versicolor and Lentinula edodes. In section 3.4 it indicates that there are two methods by which glucans have been quantified, but according to section 3.4.2 it is on Congo red, however, it includes the information of the other method, in addition, there is no section 3.4.1. review and correct.
The author could consider mentioning the extraction methods since there are also several techniques that have been used depending on the source obtained. And author doesn't mention anything about glucans being obtained from liquid culture and there are several reports indicating extraction from the culture broth.
Author Response
Please see the attachment
Rev 2. opinion
The review presents relevant information, it is coherently organized and emphasized the most relevant that has been published on glycans. The author mentions the polysaccharide of some mushrooms, however, the information he puts on pleurotan and schizophyllan is scant compared to the others (lentinan, maitake, krestin).
In general, I suggest reviewing the format, some comments about it:
In the text there is a font of different sizes (lines 7-14), the information in section 3.3 is essential, I don't see why putting smaller fonts (lines 323-349).
Reflection: The text is in font size 10, the literature in font size 9. Editors can change this scaling at any time.
2. In several parts of the text the author opens parentheses and they do not close, you should consider putting brackets in the places where there will be double parentheses
Reflection: I checked the parentheses, corrected them as necessary, and used the / / parentheses in one or two cases.
In the words that you want to abbreviate, you usually put complete and in parenthesis the abbreviation (line 141), and in line 153, put the abbreviation in parentheses, review and correct where applicable.
Reflections: I clarified the logic of abbreviations: when something is mentioned for the first time, I write it out in full, followed by the abbreviation in parentheses. I tried to do these consistently. It is not an easy question: which abbreviations are considered so well known that it is not necessary to give an explanatory explanation. One such example is UDPG, which I think should be well known. At reviewer 3's request, I will still provide the full name of the compound. In the case of UTP and UDP, I think that knowledge of these is expected from the readership.
4. Several paragraphs are not justified
Reflections: Agreeing with the comment, I eliminated quite a few paragraphs.
5.The general format of the writing is to put a title and give a space, it does not do so in section 3.6.1 (line 460)
6. Some scientific names are not in italics.
Reflection: The corrections were made.
7.The information about the glucans of Schizophyllum and Pleurotus presented is little, compared to that presented for Grifola frondosa, Trametes versicolor and Lentinula edodes.
Reflection: I have expanded the sections on pleuran and schizophyllan, using additional literature.
8.In section 3.4 it indicates that there are two methods by which glucans have been quantified, but according to section 3.4.2 it is on Congo red, however, it includes the information of the other method, in addition, there is no section 3.4.1. review and correct.
Reflections: In the original manuscript the point 3.4.1. contains the enzymatic method, the point 3.4.2. contains the Congo Red method, so I can't really understand the request.
9.The author could consider mentioning the extraction methods since there are also several techniques that have been used depending on the source obtained. And author doesn't mention anything about glucans being obtained from liquid culture and there are several reports indicating extraction from the culture broth.
Reflections: A new point (3.2.7.) was written and inserted on extraction of glucans.
General comments for all reviewers:
- I tried to correct the text based on aspects of linguistic correctness (in some places, I condense it, I broke several long sentences into parts). I tried to avoid, for example, repetition of words by using synonymous words.
- Due to the insertions and the use of a larger font size, the line numbers of review have changed therefore the number of lines given in the reviewer's opinion are no longer valid.
Finally, I ask you to approve the implemented modifications and corrections.
Submission Date
13 January 2023
Date of this review
27 Jan 2023 04:17:36

Reviewer 3 Report
In this manuscript, the authors have reviewed the the glucans of lentinan (from Lentinula edodes), pleuran (from Pleurotus ostreatus), glucans from Grifola frondosa, schizophyllan (from Schizophyllum commune), and krestin (from Trametes versicolor), along with their main biological effects. Overall, this review is well well organized and comprehensively described. However, some issues should be considered.
(1) The anstract was not well written, please rewritten it to better summarize the manuscript.
(2) Please write the full name when abbreviation first represented in the manuscript, like DM, UDP, UDPG, etc.
(3) Fig. 1 was a screenshot and not well presented, please redrawn this figure.
(4) On line 117, which meaning do “their” refer to? Please clarify it. Line 110-114 and line 117-127 need to be integrated into one passage.
(5) Line 128-131, please add the reference.
(6) Table 2 only listed source of β-glucan, please add more examples from other sources.
(7) Table 3 can be removed with its content described in line 377-380, or integrated with Table 4.
(8)The data of cultivated taxa in Table 6 was not well presented, please add the “±”.
(9) Table 7 was repeated presented in the manuscript.
(10) Some minor language problems need to be modified, like line 152, „Starch”, line 334 “spelling”, etc.
Author Response
Please see the attachment
Rev 3 opinion
n this manuscript, the authors have reviewed the the glucans of lentinan (from Lentinula edodes), pleuran (from Pleurotus ostreatus), glucans from Grifola frondosa, schizophyllan (from Schizophyllum commune), and krestin (from Trametes versicolor), along with their main biological effects. Overall, this review is well organized and comprehensively described. However, some issues should be considered.
- The abstract was not well written, please rewritten it to better summarize the manuscript.
Reflections:
The work is of an overview and summary (review) nature and aims at a multi-faceted presentation of mushroom glucans (from chemistry to use). In this case, the abstract can be understood as a "table of contents" that draws attention to the topics to be presented. I made some minor changes in the original text. The other two reviewers did not make any suggestions to change the abstract. Based on this, I respectfully ask you to accept the content of the abstract.
- Please write the full name when abbreviation first represented in the manuscript, like DM, UDP, UDPG, etc.
Reflections: At your request, I added full name (abbreviated name) pairs. However, it is not always easy to decide what is known and what is not. In the case of the UDPG, I entered the full name, but not in the case of the UTP and UDP, since the expected professional qualifications of those reading the work may make these unnecessary.
- 1 was a screenshot and not well presented, please redrawn this figure.
Reflections: I created the figure in the "Power Point" system, then reformatted it into a figure. In terms of content, it clearly presents the main groups and types of mushroom polysaccharides with examples. Given that the other two reviewers of the thesis did not criticize the figure, I respectfully ask the reviewer Colleague to accept the way I present the mushroom carbohydrates in the figure.
- On line 117, which meaning do “their” refer to? Please clarify it. Line 110-114 and line 117-127 need to be integrated into one passage.
Reflection: a. The word "their" refers to the sentence before the figure, i.e. monosaccharides. To avoid misunderstandings, I re-edited the given section, the monosaccharide sentence is placed after the figure.
- I combined the requested contents into one paragraph.
(5) Line 128-131, please add the reference.
Reflection: The requested citation: the book of Kalac [6]
- Table 2 only listed source of β-glucan, please add more examples from other sources.
Reflection: Some of the new data in the supplemented table show the very diverse molecular weight of glucans of various origins and types, and in addition, different methods were used for these as well.
(7) Table 3 can be removed with its content described in line 377-380, or integrated with Table 4.
Reflections: Table 3 compares the glucan contents of 18 wild mushroom species obtained by two methods. Table 4 compares the glucan contents of the three most important cultivated species. I think that combining the two tables would not improve the presentation of what is being said. Therefore, please accept the independent existence of tables 3 and 4.
(8)The data of cultivated taxa in Table 6 was not well presented, please add the “±”.
Reflections: I have added to the table, the SD is included for all basic data, as well as the SD value obtained during the meta-analysis.
(9) Table 7 was repeated presented in the manuscript.
Reflection: I have removed the table.
(10) Some minor language problems need to be modified, like line 152, „Starch”, line 334 “spelling”, etc.
Reflections: I have corrected the indicated errors.
General comments for all reviewers:
- I tried to correct the text based on aspects of linguistic correctness (in some places, I condense it, I broke several long sentences into parts). I tried to avoid, for example, repetition of words by using synonymous words.
- Due to the insertions and the use of a larger font size, the line numbers of review have changed therefore the number of lines given in the reviewer's opinion are no longer valid.
Finally, I ask you to approve the implemented modifications and corrections.
Submission Date
13 January 2023
Date of this review
07 Feb 2023 04:51:58

Round 2
Reviewer 1 Report
There are some small correction and suggestion describing below.
Line 87: total carbohydrate content (%) = [100- (moisture + crude protein + crude lipid + ash + crude fibre)
line 92: high molecular weight polysaccharide
line 251: HPGPC: high performance gel permeation chromatography
GPV: gel permeation chromatography
NMR: nuclear magnetic resonance
Line 260: the biological effect of glucans with a higher molecular weight is usually greater, (possible coming from multivalency effects?)
About identical references format including journal abbreviation, etc, , the publishing house should help authors to deal with it.
Author Response
Rev. 1. (second round)
There are some small correction and suggestion describing below.
- Line 87: total carbohydrate content (%) = [100- (moisture + crude protein + crude lipid + ash + crude fibre)
Reflection: Yes, I added the missing word.
- line 92: high molecular weight polysaccharide
Reflection: I made the correction
- line 251: HPGPC: high performance gel permeation chromatography
GPV: gel permeation chromatography
NMR: nuclear magnetic resonance
Reflection: I corrected the uppercase letters to lowercase.
- Line 260: the biological effect of glucans with a higher molecular weight is usually greater, (possible coming from multivalency effects?)
Reflection: The literature does not attempt to explain this phenomenon, so I did not wish to venture into hypotheses either. A multivalency effect is possible, but I found no evidence for it.
- About identical references format including journal abbreviation, etc, , the publishing house should help authors to deal with it.
Reflection: In the references section, I changed the names of the journals to abbreviated
forms.
Please accept the corrections made. Thank you.
The author
Submission Date
13 January 2023
Date of this review
15 Feb 2023 02:57:55
